# Canonical androgen response element motifs are tumor suppressive regulatory elements in the prostate

Xuanrong Chen [1,2,10], Michael A. Augello[1,2,10], Deli Liu[1,2,3], Kevin Lin[2], Alex Hakansson[4], Martin Sjöström [5], Francesca Khani [6], Lesa D. Deonarine[2], Yang Liu [4], Jaida Travascio-Green[2], Jiansheng Wu[1,2], Un In Chan [2], Jude Owiredu[2,7], Massimo Loda [2,6], Felix Y. Feng [5,8], Brian D. Robinson [2,6,9], Elai Davicioni[4], Andrea Sboner [3,6,9] & Christopher E. Barbieri [1,2,9] ✉

The androgen receptor (AR) is central in prostate tissue identity and differentiation, and controls normal growth-suppressive, prostate-specific gene expression. It also drives prostate tumorigenesis when hijacked for oncogenic transcription. The execution of growth-suppressive AR transcriptional programs in prostate cancer (PCa) and the potential for reactivation remain unclear. Here, we use a genome-wide approach to modulate canonical androgen response element (ARE) motifs—the classic DNA binding elements for AR—to delineate distinct AR transcriptional programs. We find that activating these AREs promotes differentiation and growth-suppressive transcription, potentially leading to AR⁺ PCa cell death, while ARE repression is tolerated by PCa cells but deleterious to normal prostate cells. Gene signatures driven by ARE activity correlate with improved prognosis and luminal phenotypes in PCa patients. Canonical AREs maintain a normal, lineage-specific transcriptional program that can be reengaged in PCa cells, offering therapeutic potential and clinical relevance.

Many cancers show dependence on tissue and lineage-specific transcription factors that are critical for normal tissue as well[1]. The AR is the central determinant of prostate tissue identity, lineage, and differentiation, controlling normal, and growth-suppressive prostate-specific gene expression[2]. However, it is also a key driver of prostate tumorigenesis, becoming "hijacked" through epigenetic reprogramming to drive oncogenic transcription[3–6]. Importantly, it remains the key therapeutic target for PCa, even in advanced, treatment-resistant disease[7], where genomic alterations such as AR gene and regulatory

element amplification, overexpression, mutations, and splice variants of AR drive continued reliance on androgen signaling[8].

Most focus in the field of AR reprogramming has been on the gain of oncogenic functions by AR, associated with a shift in cistromic localization and control of new target genes associated with oncogenic effects such as proliferation and invasion[3,9]. However, reprogramming in both directions may be important—loss of tumor suppressive functions may be critical. Importantly, evidence suggests oncogenic and growth-suppressive transcriptional programs controlled by AR are

[1]Department of Urology, Weill Cornell Medicine, New York, NY, USA. [2]Sandra and Edward Meyer Cancer Center, Weill Cornell Medicine, New York, NY, USA. [3]The HRH Prince Alwaleed Bin Talal Bin Abdulaziz Alsaud Institute for Computational Biomedicine, Weill Cornell Medicine, New York, NY, USA. [4]Veracyte, Inc., South San Francisco, CA, USA. [5]Department of Radiation Oncology, University of California, San Francisco, CA, USA. [6]Department of Pathology and Laboratory Medicine, Weill Cornell Medicine, New York, NY, USA. [7]Biochemistry, Cellular, Developmental and Molecular Biology Program, Weill Cornell Medicine, New York, NY, USA. [8]Departments of Urology and Medicine, University of California, San Francisco, CA, USA. [9]Caryl and Israel Englander Institute for Precision Medicine, Weill Cornell Medicine, New York, NY, USA. [10]These authors contributed equally: Xuanrong Chen, Michael A. Augello. ✉e-mail: chb9074@med.cornell.edu

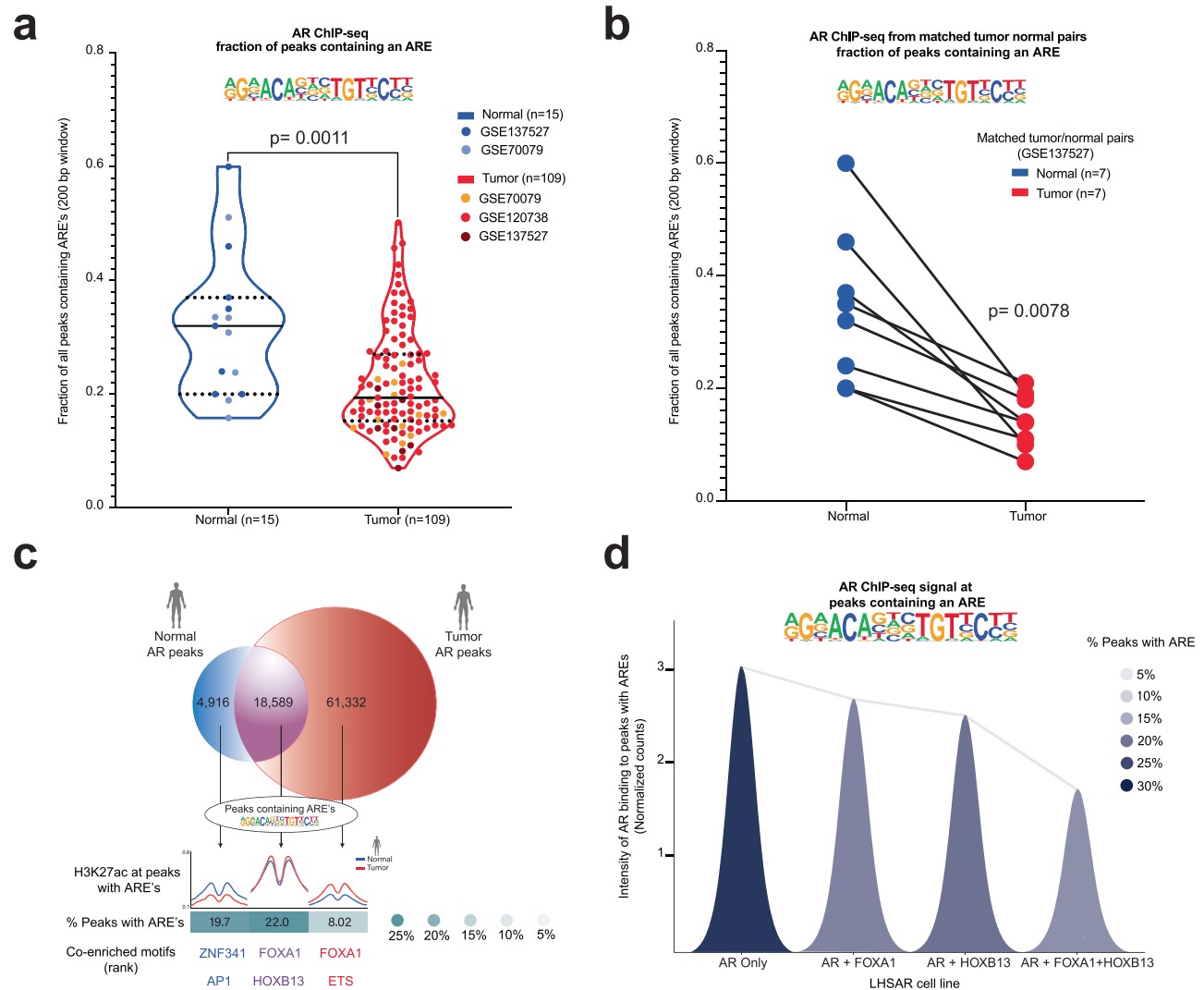

**Fig. 1 | Canonical AR Response Element (ARE) motifs are depleted in human prostate cancer. a** Fraction of peaks containing an ARE from AR ChIP-seq data in normal human prostate tissue (blue) and PCa tissue (red). *P* = 0.0011 by Mann-Whitney two-sided test. **b** Fraction of peaks with ARE from AR ChIP-seq data in matched normal human prostate tissue (blue) and PCa (red) from the same patients. *P* = 0.0078 by Wilcoxon two-sided test. **c** Overlap of collated AR peaks from normal and tumor prostate samples, with associated enhancer activity (H3K27ac) at normal specific (blue), tumor specific (red), and common (purple) AR peaks, along with percent containing AREs, and other associated motifs. **d** Intensity of AR binding in peaks with an ARE in LHSAR cells expressing AR alone or associated oncogenic factors (FOXA1 and HOXB13). AR binding to peaks with AREs decreases with the addition of FOXA1 and HOXB13. Source data are provided as a Source Data file.

associated with distinct epigenomic regulation. Classically, nearly all AR-regulated gene expression is mediated through direct AR binding to palindromic DNA sequences known as Androgen Response Elements (AREs)[2]. In PCa, the AR cistrome is distinct and highly associated with motifs for FOXA1 and HOXB13, and these can drive the AR cistrome to reflect PCa[3,10–12], while other genomic alterations impair the normal growth suppressive cistrome and transcriptome of AR[5], revealing the plasticity and context-dependency of AR programs.

However, several major issues remain largely unexplained: What is the importance of canonical AREs in differentiating oncogenic and growth-suppressive AR-driven transcriptional programs? Can these programs be separated and independently modulated? Do canonical AREs remain essential for PCa cells? We, therefore, sought to better characterize the epigenomic regulation of the growth suppressive AR program, its control by AREs, and its deregulation in human PCa.

In this work, we employ a genome-wide strategy to modulate regulatory elements containing AREs to define distinct AR transcriptional programs. We demonstrate that canonical AREs are responsible for a normal, growth-suppressive, lineage-specific transcriptional program, that this can be reengaged in PCa cells for potential therapeutic benefit, and genes controlled by this mechanism are clinically relevant in human PCa patients.

## Results

### AREs are enriched in the AR cistrome of normal prostate tissue and depleted in prostate cancer

In cohorts of human normal prostate tissue and PCa samples[3,4,13], AREs were less common in the AR cistrome of PCa compared to normal prostate tissue (Fig. 1a). Further interrogation in a dataset of matched tumor and normal tissue demonstrated that each paired patient sample showed depletion of AR binding near AREs in tumor compared to matched normal[13] (Fig. 1b). Enhancer activity (H3K27ac ChIP-seq) confirms that AR binding patterns reflect active regulatory control at tumor- and normal-specific AR peaks (Fig. 1c). Moreover, ectopic expression of AR reprograming factors associated with PCa (FOXA1 and HOXB13) in non-transformed prostate cells[3] and mouse normal prostate cells[10] resulted in depletion of AREs from the AR cistrome (Fig. 1d and Supplementary Fig. 1a). Finally, in PCa

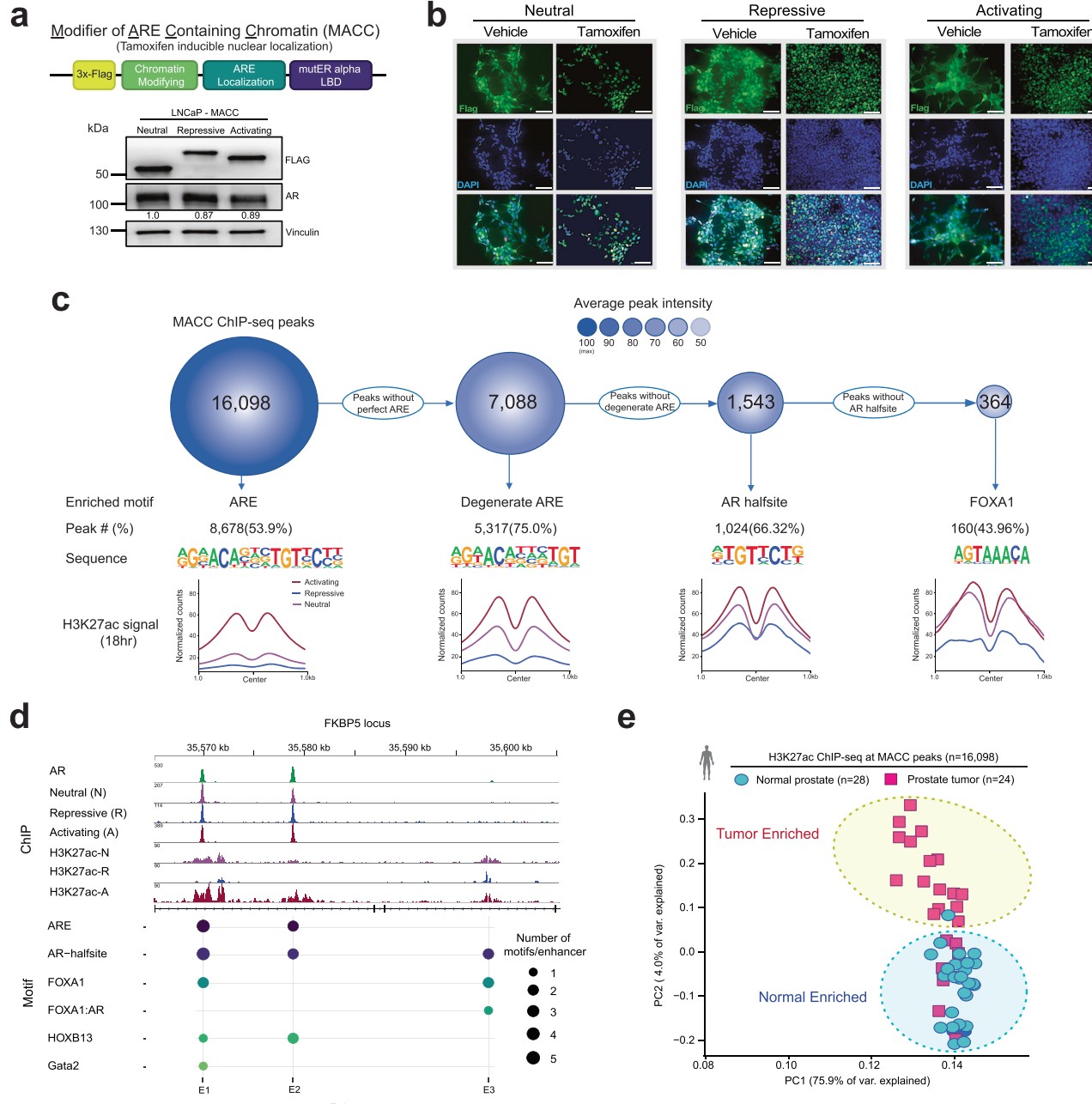

**Fig. 2 | MACCs represent an inducible system to directly modulate ARE-containing regulatory elements. a** Schematic of MACC constructs, and expression of neutral (3X FLAG only), repressive (KRAB) and activating (VP64) constructs. The samples derive from the same experiment, but different gels for Flag, AR, and Vinculin were processed in parallel. The experiment was repeated three times (biological replicates), and a representative example is shown here. **b** Nuclear localization of constructs upon tamoxifen induction. Scale bar = 20 μm. The experiment was repeated 3 times (biological replicates), and a representative example is shown here. **c** ChIP-seq of MACC constructs in LNCaP cells, showing maximal binding and effect on enhancer activity at ARE motifs, with decreasing affinity and effect on H3K27ac activity for associated motifs. **d** Example of MACC localization and modulation of regulatory activity at the FKBP5 locus, with ARE-containing enhancers 1 and 2 (E1 and E2) affected by MACCs, but E3 (without an ARE) is insensitive. **e** H3K27ac signal at MACC peaks distinguishes tumor from normal human prostate tissue. For (**a**, **b**) experiments were conducted at least three times with consistent results. Source data are provided as a Source Data file.

models[14,15], overexpression of FOXA1 also resulted in the depletion of AREs from the AR cistrome, while depletion of FOXA1 had the opposite effect (Supplementary Fig. 1b–g). Therefore, the AR cistrome was associated with ARE motifs in normal prostate tissue and de-enriched in PCa, suggesting a tumor-suppressive regulatory element. We therefore hypothesized that activation of ARE-containing regulatory elements would have tumor suppressive effects in prostate cells.

## Development and validation of a strategy to modulate ARE-associated regulatory elements

To test this, we developed inducible constructs to modulate chromatin activity around AREs. *MACCs* (Modifiers of ARE Containing Chromatin – Fig. 2a) are designed to localize to AREs via the DNA binding domain of AR, but lack the N-terminal region largely responsible for recruitment of co-factors[16] ("Methods" section and Supplementary Fig. 2). MACCs are tamoxifen-inducible, epitope-tagged (3X FLAG), and

contain repressive or activating chromatin modifying domains (H3K9 methyltransferase KRAB, or transcriptional activator VP64). We examined localization and activity of three different MACCs, named regarding expected transcriptional effects: (1) Neutral (**N**, 3X FLAG alone, no chromatin modifying domain), (2) Repressive (**R**, KRAB), and (3) Activating (**A**, VP64). In stable LNCaP lines, after confirming expression and tight inducible nuclear localization with tamoxifen (Fig. 2a, b), we examined genome-wide distribution with ChIP-seq. Consistent with appropriate localization, consensus MACC peaks (Supplementary Fig. 3a, b) localized primarily to introns and intergenic regions (Supplementary Fig. 3c), and the canonical ARE was the most significantly enriched motif, followed by lower affinity variants of the ARE such as AR-halfsites[17,18] (Fig. 2c). Modulation of H3K27ac was consistent with expected activity−at ARE-containing peaks, H3K27ac was high with activating MACC, and lowest with repressive (Fig. 2c and Supplementary Fig. 3d). The well-characterized AR-target gene *FKBP5* provides a clear example, with three enhancer elements (E1, E2, and E3) bound by AR[19]. E1 and E2 contain AREs, are bound by MACC constructs, and show expected modulation of H3K27ac, while E3 lacks a canonical ARE motif, and shows no MACC effects on H3K27ac (Fig. 2d). Further, MACC activity was restricted to AR-bound chromatin enriched for AREs (Supplementary Fig. 3e), as other AR binding sites lacking an ARE showed no global difference in H3K27ac signal (Supplementary Fig. 3e). Finally, interrogation of H3K27ac ChIP-seq data from human patient samples[20] showed that enhancer activity at MACC consensus peaks distinguishes tumor from normal tissue (Fig. 2e), highlighting the relevance of the binding sites of these engineered constructs in human prostate tissue. These MACC constructs represent tools to modulate ARE-containing regulatory elements, reveal and manipulate distinct subsets of AR-responsive enhancers, and interrogate cancer and normal-specific signaling pathways, with clear applicability to human specimens.

## Activation of ARE-enriched enhancers is growth suppressive in prostate cancer cells in vitro and in vivo and dispensable for tumorigenic phenotypes

To determine the biological impact of directly modulating these ARE-containing regulatory elements, we examined the effect of induction of MACC constructs in androgen-dependent PCa cell lines. Dogma states that PCa cells respond to AR activation with concomitant increased expression of AR target genes and increased proliferation[21], and that these are tightly coupled. To clarify the effects on the transcriptional programs controlled specifically by AREs in PCa cells, we performed gene expression profiling with RNA-seq in LNCaP with tamoxifen-inducible activation of MACC constructs (Fig. 3a, b). Principle component analysis showed that without induction (vehicle) alone, all LNCaP lines had similar transcriptomes (Fig. 3a). In contrast, induction with tamoxifen led to dramatic changes in transcriptomes with activating and repressive MACCs (Fig. 3a). Pathway analysis showed that activation of AREs resulted in upregulation of classic AR target genes (GSEA AR Hallmarks[22]), with simultaneous down-regulation of gene sets associated with proliferation (Fig. 3b). These results were further confirmed by analyzing clinically relevant transcriptional signatures of AR (AR score)[23] and the cell cycle (RB loss signature)[24] (Fig. 3c). Activation of AREs was associated with a higher AR score but was de-enriched cell cycle activity. Conversely, ARE repression resulted in an inverted signature, with a lower AR score and higher cell cycle de-regulation signature (Fig. 3b, c). Collectively, these data suggest that there is a decoupling of AR activity from cell cycle effects by modulating enhancers enriched for AREs.

Phenotypically, activation of AREs resulted in severe growth suppression and cell death in both basal growth conditions and androgen-simulated growth after androgen starvation (Fig. 3d and Supplementary Fig. 4). Repression of AREs had minimal effect in 2D culture (Fig. 3d and Supplementary Fig. 4), but stimulated increased

growth of LNCaP cells as 3D spheroids (Fig. 3e). Similar effects were observed in other androgen-dependent prostate cell lines (LAPC4) and LNCaP/AR cells with higher AR expression levels (Supplementary Fig. 5a–d), but non-prostate cells (293T) and androgen-indifferent PCa cells (PC3, DU145) showed no effect of MACC induction (Supplementary Fig. 5e-g), consistent with lineage specificity. To confirm that effects were not artifact related to hormonal treatment with tamoxifen, we engineered an additional doxycycline-inducible system with the same biological effects (Supplementary Fig. 5a). Finally, activation of AREs in vivo also suppressed the growth of LNCaP xenografts in nude mice, while ARE repression was dispensable for tumor growth (Fig. 3e and Supplementary Fig. 6). Together, these data show that direct activation of regulatory elements containing AREs results in growth suppression and cell death of PCa cells.

## Repression of ARE-associated chromatin disrupts differentiation in normal prostate epithelial cells

Our data in PCa cell lines showed that activation of AREs results in growth suppressive phenotypes, while repression of AREs has minimal effects. We next sought to determine the effects of manipulating these regulatory elements in normal prostate epithelial cells by deploying inducible MACC constructs in genetically normal mouse prostate organoids. Neutral and activating MACCs had no distinguishable effect on organoid phenotypes, while the repressive MACC resulted in severe disruption of growth and luminal features in two independent mouse organoid lines (Fig. 4a, b and Supplementary Fig. 7a–d). In direct contrast to PCa cell lines, repression of AREs in human benign prostate epithelial cells (RWPE1) resulted in growth suppression, while activation of AREs had fewer effects (Fig. 4c and Supplementary Fig. 7e, f). Gene expression profiling in these non-transformed prostate cells was consistent with these effects, with downregulation of gene sets associated with proliferation (e.g. Myc and E2F targets, G2M checkpoint) and ARE repression (Fig. 4d, Supplementary Fig. 8a, b). In addition, modulation of AREs revealed regulation of prostate epithelial differentiation phenotypes. In both LNCaP and RWPE1 cells, ARE repression led to downregulation of genes associated with luminal epithelia (LE), and upregulation of basal epithelial (BE) markers, while ARE activation maintained luminal gene expression (Fig. 4e and Supplementary Fig. 8c, d), minor changes in stem-like, neuroendocrine (NE), and epithelial-mesenchymal transition (EMT) pathways (Supplementary Fig. 8c, d). Finally, in normal mouse prostate organoids, immunofluorescence showed altered expression of luminal (cytokeratin 8) and basal (cytokeratin 5) markers with ARE modulation (Fig. 4f). These data show that in normal prostate epithelial cells, elevation of transcriptional programs controlled by AREs has minimal effects, while disruption of these regulatory elements results in growth suppression and loss of luminal epithelial phenotypes.

## HDAC3 restricts the growth-suppressive effect by binding to the ARE-enriched enhancers

To elucidate how direct modulation of regulatory elements containing AREs shapes the chromatin states to regulate oncogenic and normal-specific signaling pathways, we first examined the genomic features of MACC H3K27ac peaks by classifying them into six categories based on their co-occupancy status and association with chromatin modifications. Using ChromHMM[25] with six distinct histone marks (H3K4me1, H3K4me2, H3K4me3, H3K27me3, H3K36me3, H3K79me2) under androgen stimulation, we defined various chromatin states associated with MACCs (Fig. 5a), ranging from heterochromatin (E1) to active enhancers/promoters (E6; see Methods for detail). MACCs exhibited the expected dynamic changes in active enhancers/promoters at the E6 state, with either repressive or activating chromatin-modifying domains. The enrichment of the E5 state indicates significant chromatin remodeling at bivalent enhancers/promoters, consistent with AR's established role in regulating tumorigenesis/differentiation

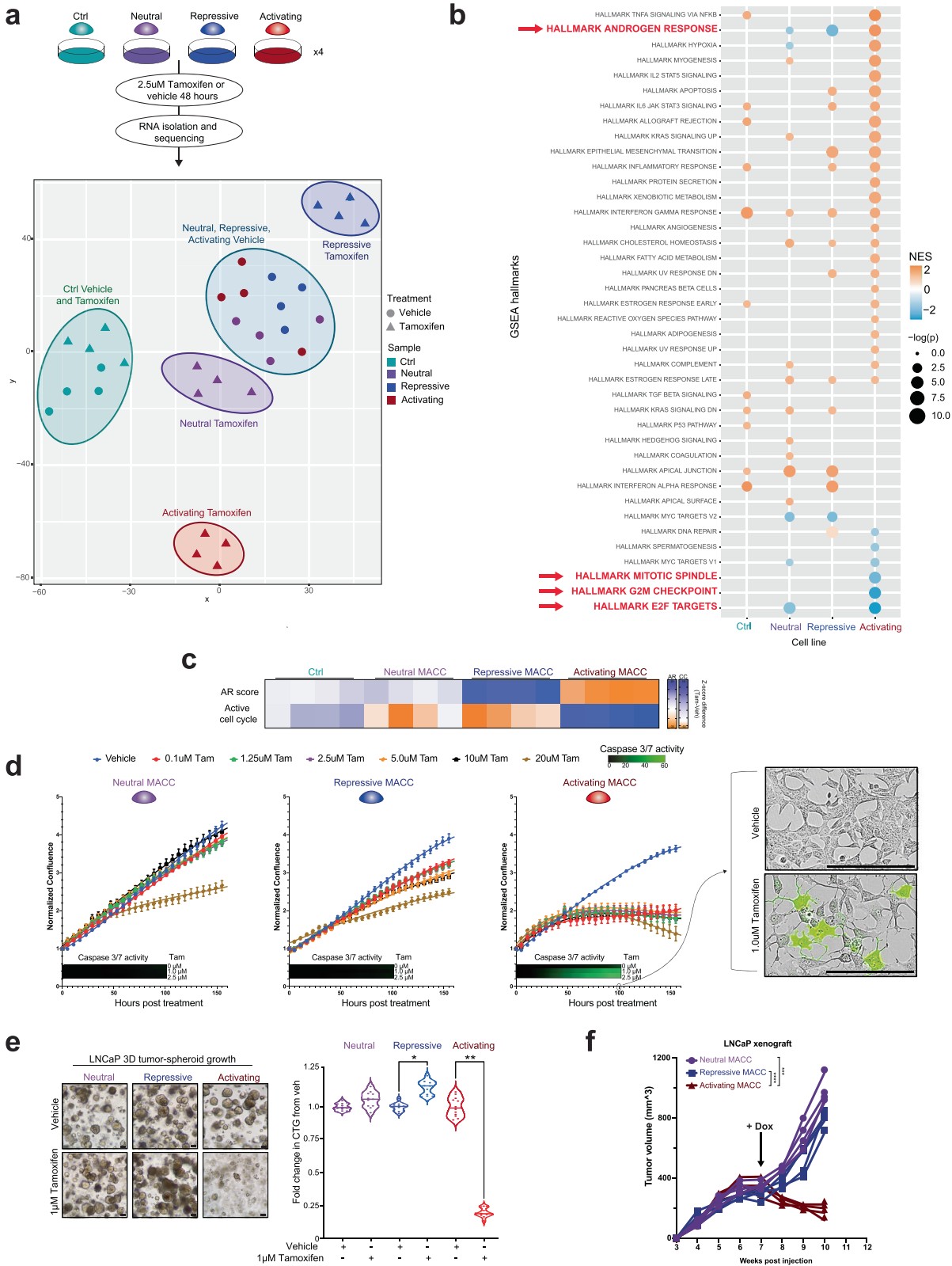

through bivalent enhancer[26–28]. We then focused on histone deacetylases (HDACs) and EZH2, given their common overexpression and bivalent transcriptional function in cancers[26,29–31]. By mapping the binding sites of these factors to MACC H3K27ac peaks, it was evident that only HDAC3, and not HDAC1/2 or EZH2, showed a dynamic response after androgen stimulation[30] (Fig. 5b and Supplementary Fig. 9). Furthermore, HDAC3 binding sites were also associated with MACC

peaks (Supplementary Fig. 10a) and were restricted to AR-bound chromatin enriched for AREs (Supplementary Fig. 10b), as other AR binding sites lacking an ARE showed no global difference in HDAC3 signal after androgen stimulation (Supplementary Fig. 10b). To determine HDAC3's effect on ARE-associated chromatin, we analyzed motif enrichment at HDAC3 binding sites (HDAC3 ChIP-seq)[30] before and after androgen stimulation and found ARE motifs were

**Fig. 3 | Modulation of AREs results in uncoupling of canonical AR target genes and proliferation in prostate cancer cells, with ARE activation growth suppressive, while ARE repression is tolerated. a** Experimental plan for transcriptional profiling, and PCA of gene expression profiled from vehicle or tamoxifen-treated LNCaP MACC lines. **b** Pathway analysis using GSEA Hallmarks of induced vs. vehicle LNCaP MACC lines. The enrichment analysis was generated using Fisher's exact test. **c** Opposing effects on AR target genes (AR score) and cell cycle genes (RB loss signature) with activation or repression of AREs. **d** Growth of LNCaP MACC lines in 2D culture with vehicle or increasing doses of tamoxifen. Neutral and repressive constructs have minimal effect; Activating MACC is growth suppressive

and induces apoptosis as measured by Caspase 3/7 activity. Scale bar = 20 μm, n = 4. Data are shown as mean ± SD as representative results from three independent experiments. **e** Brightfield images and growth of LNCaP MACC lines as 3D spheroids, +/- tamoxifen. Scale bar = 20 μm, n = 12 for all conditions, two-tailed Student's t-test, *p < 0.05, **p < 0.01. The experiment was repeated 3 times (biological replicates), and a representative example is shown here. **f** LNCaP xenografts with doxycycline-inducible MACC constructs in nude mice (4 mice per group). Doxycycline chow was started at week 7. Two-way ANOVA, ***p < 0.001, ****p < 0.0001. Source data are provided as a Source Data file.

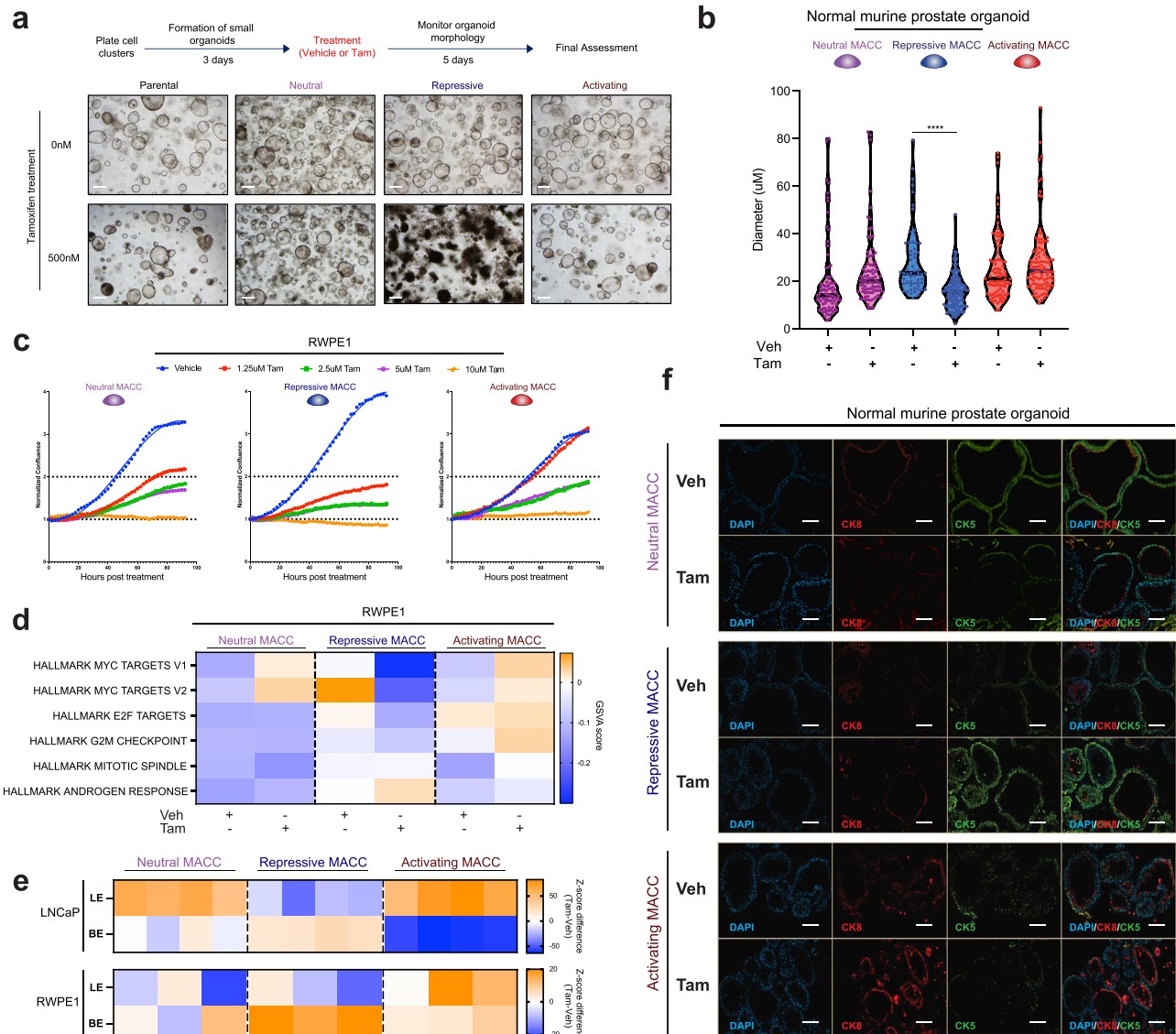

**Fig. 4 | In benign prostate cells, ARE activation is well tolerated, but ARE repression results in altered growth and differentiation. a, b** Genetically normal mouse prostate organoids (**a**, #1 and **b**, #2) with inducible MACC constructs, showing loss of luminal morphology with repression of AREs. Scale bar = 40 μm. Two-tailed Student's t test, ****p < 0.0001. Each point represents a separate organoid cell. **c** Growth of benign RWPE1 prostate MACC lines with vehicle or increasing doses of tamoxifen, n = 3. **d** Pathway analysis using gene expression profiling of induced vs. vehicle RWPE1 MACC lines. **e** Heatmap of LE and BE markers' gene

signature levels in LNCaP and RWPE1 MACC lines. **f** Representative immunofluorescence images of the expression of luminal (cytokeratin 8) and basal (cytokeratin 5) markers in normal mouse prostate organoids (#2). Organoids were treated with either vehicle or 2 μM Tamoxifen for 48 h. Scale bar = 20 μm. The experiment was repeated three times (biological replicates), and a representative example is shown here. For (**a–c**, **f**) experiments were conducted at least three times with consistent results. Source data are provided as a Source Data file.

significantly enriched (Fig. 5c and Supplementary Fig. 10c, d), unlike FOXA1 or HOXB13 motifs (Supplementary Fig. 10d). This suggests that HDAC3 plays a role in modulating the activation of those regulatory elements containing AREs in the prostate and HDAC3 activity could

lead to alterations in the growth-suppressive phenotypes upon ARE activation. We then knocked out HDAC3 in our LNCaP ARE activation cells, which indeed intensified the growth-suppressive phenotypes upon ARE activation (Fig. 5d, e). To further confirm that HDAC3

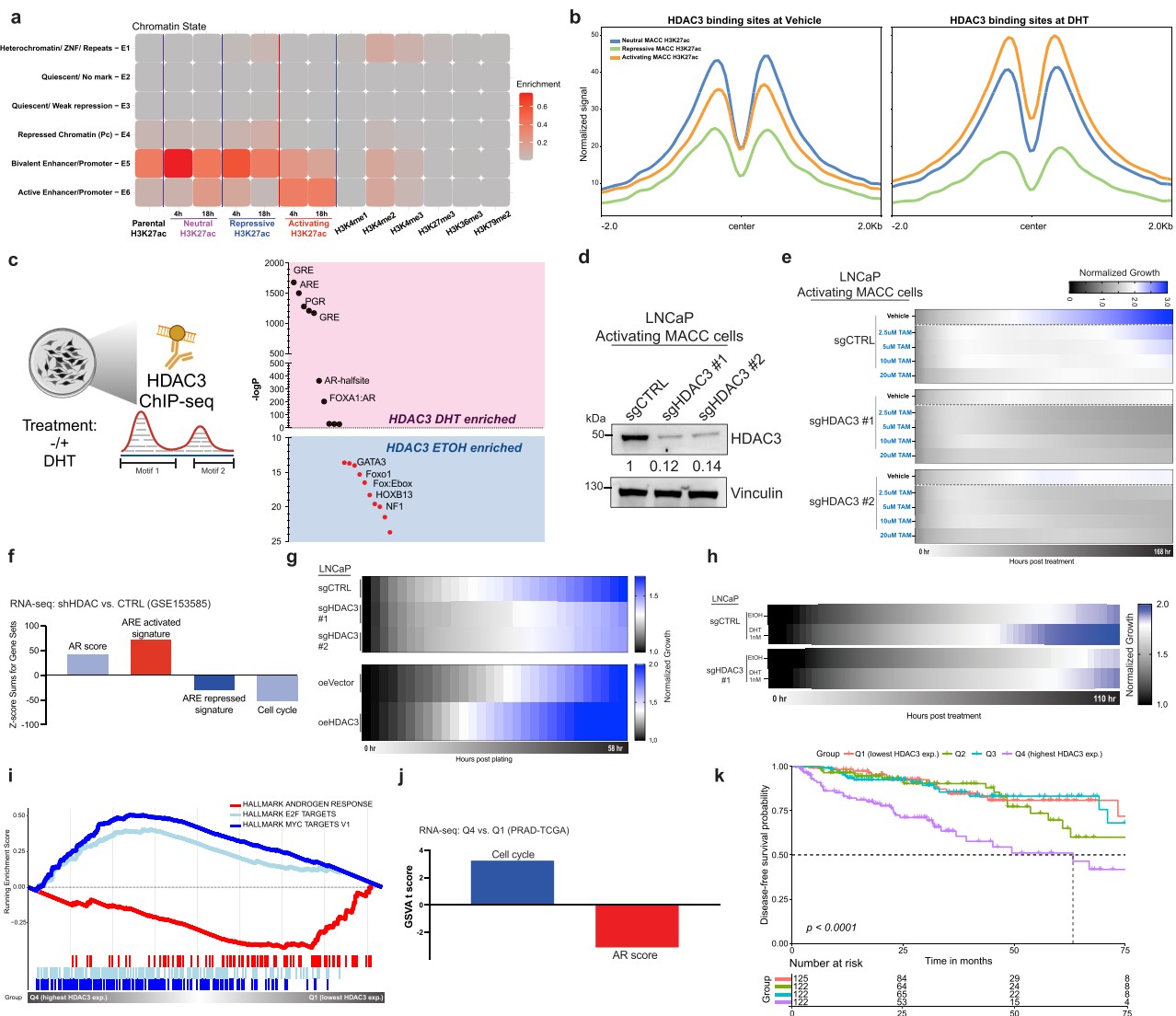

**Fig. 5 | HDAC3 mitigates the growth-suppressive effect of AREs. a** Combinatorial pattern of histone marks in a 6-state model using ChromHMM. The heatmap (Emission plot) displays the frequency of the six distinct histone modifications (H3K4me1—active enhancer, H3K4me2, H3K4me3—active promoter, H3K27me3—repressive epigenetic mark, H3K36me3—transcription mark, H3K79me2) under androgen stimulation along with MACCs in each state. **b** Signal of MACC H3K27ac (18 h) at all HDAC3 vehicle-treated (GSM717402) or HDAC3 DHT-treated (GSM717403) binding peaks. **c** Schematic of motif enrichment analysis from HDAC3 ChIP-seq before and after DHT stimulation. ARE motifs were enriched in the DHT condition at HDAC3 binding sites. **d** Immunoblot of HDAC3 expression in LNCaP activating MACC cells with control or two independent HDAC3 sgRNAs. The samples derive from the same experiment, but different gels for HDAC3 and Vinculin were processed in parallel. **e** Growth of HDAC3 knockout in LNCaP activating MACC cells with vehicle or increasing doses of tamoxifen. HDAC3 knockout amplified the

growth-suppressive phenotypes upon ARE activation. The growth readout is presented in a heatmap format, *n* = 3. **f** Pathway analysis using gene expression profiling of HDAC3-knockdown RNA-seq data in LNCaP cells (GSE153585). **g** Growth of HDAC3 knockout or overexpression in LNCaP cells. The growth readout is presented in a heatmap format, *n* = 3. **h** Growth of LNCaP HDAC3 knockout cells after androgen deprivation and DHT stimulation. The growth readout is presented in a heatmap format, *n* = 3. **i, j** Pathway analysis using GSEA Hallmarks and gene signatures of Q4 vs. Q1 groups based on HDAC3 expression level in the TCGA cohort. **k** Kaplan–Meier analysis of disease-free survival in the TCGA cohort, stratified into quartiles by HDAC3 expression level. Time = months. Statistical significance was determined using the log-rank test. For (**d, e, g, h**) experiments were conducted at least three times with consistent results. Source data are provided as a Source Data file. **c** Created with BioRender.com.

restricts the ARE activation program, we applied transcriptional signatures from each specific MACC construct, AR- and cell cycle scores to RNA-seq profiles generated upon HDAC3 kockdown[12]. Disruption of HDAC3 expression correlated with a higher AR score and increased ARE activation activity, while cell cycle activity and ARE repression activity were diminished (Fig. 5f and Supplementary Fig. 10e, f). Accordingly, this resulted in impaired cell growth in LNCaP cells, in contrast to the promotion of cell growth observed with HDAC3 over-expression (Fig. 5g and Supplementary Fig. 10e), and a reduced response to androgen-mediated growth (Fig. 5h). We further explored

the clinical relevance of these observations by employing gene set enrichment analysis on pathway hallmarks utilizing patient samples available in The Cancer Genome Atlas (TCGA)[32]. Patients (Supplementary Fig. 10g) were stratified into four groups based on their HDAC3 expression levels: Q1 (lowest HDAC3 expression), Q2, Q3, and Q4 (highest HDAC3 expression). Significantly, low HDAC3 expression correlated with elevated AR activity, as evidenced by enrichment in the canonical Androgen Response pathway and AR score (Fig. 5i, j and Supplementary Fig. 10h). Conversely, high HDAC3 expression was associated with an enrichment of cell cycle-related pathways (Fig. 5i, j

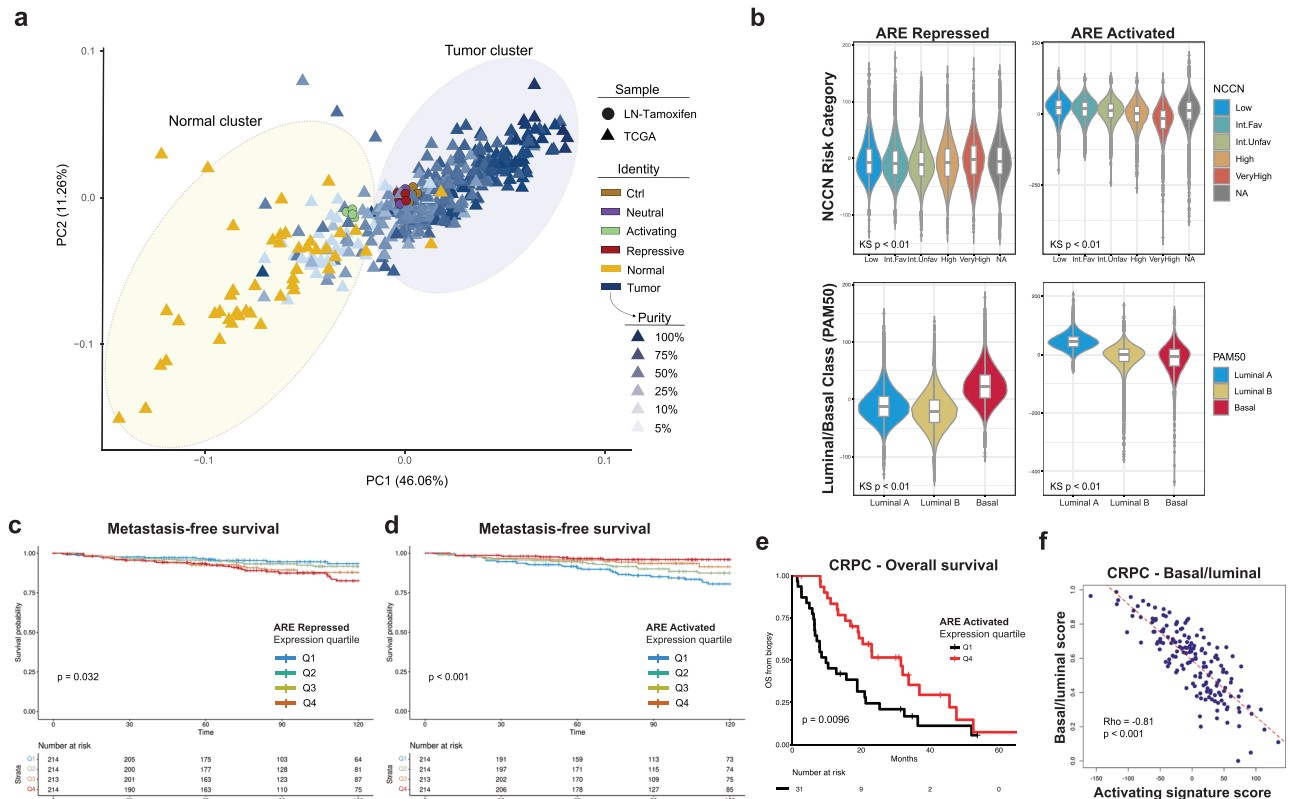

**Fig. 6 | Distinct transcriptional programs revealed by modulating AREs are clinically relevant in human prostate cancer. a** Unsupervised clustering of the transcriptomes of human prostate cancer and normal samples (TCGA), along with tamoxifen induced LNCaP MACC lines. **b** Association of ARE and activated signatures from LNCaP MACC lines with National Comprehensive Cancer Network (NCCN) risk category (top) and PAM50 luminal/basal classification (bottom) in clinically localized prostate cancer, n = 169,123 (n = number of patients; center: median; box: 25th to 75th percentile; shape: the distribution of the data, with the width indicating the kernel density estimate of the frequency). KS Kruskal–Wallis, Int.Fav Favorable Intermediate risk, Int.Unfav Unfavorable Intermediate risk, also Supplementary Table 1 and 3. **c** Kaplan–Meier analysis of metastasis-free survival in 855 men after radical prostatectomy, stratified into quartiles by ARE repressed

signature expression level. Time = months. Statistical significance was determined using the log-rank test. **d** Kaplan–Meier analysis of metastasis-free survival in 855 men after radical prostatectomy, stratified into quartiles by ARE activated signature expression level. Time = months. Statistical significance was determined using the log-rank test. **e** Kaplan–Meier analysis of overall survival in 123 men with metastatic castrate resistant prostate cancer, stratified into quartiles by ARE activated signature expression level in biopsy specimens. Hazard ratio (HR) = 0.47 [0.26–0.83]. Statistical significance was determined using the log-rank test. **f** Correlation analysis of basal/luminal phenotype score (higher = more basal) by gene expression (Y-axis) with ARE activated signature expression level (X-axis). Rho = −0.81, Spearman's correlation. Statistical significance was evaluated using a two-sided Spearman's rank correlation test.

and Supplementary Fig. 10h). Furthermore, higher levels of HDAC3 were indicative of poorer disease-free survival (DFS) (Fig. 5k). These data are consistent with the proposed regulatory function of HDAC3 in tempering the growth-inhibitory effects driven by ARE activation. Combined, we conclude that HDAC3 acts to dampen the growth-suppressive programs associated with ARE-enriched enhancers, highlighting its role in modulating AR-mediated transcriptional regulation.

**Distinct transcriptional programs revealed by modulating AREs are clinically relevant in human prostate cancer**
We next asked whether the distinct AR target genes revealed by our strategy to directly modulate AREs were relevant in human prostate samples. We examined the relationship of transcriptional programs controlled by AREs to human PCa and normal prostate tissue samples. Unsupervised clustering of the transcriptomes of LNCaP cells with inducible expression of MACCs with human PCa and normal prostate samples[32] revealed that all samples clustered with tumor samples except those with activation of AREs, whose transcriptional profile shifted toward normal prostate tissue (Fig. 6a and Supplementary Fig. 11a). Similar effects were observed in an independent cohort with matched cancer and normal tissue[33] (Supplementary Fig. 11b, c). Gene expression signatures specific to ARE activation or repression were

derived from the transcriptional responses of LNCaPs to different MACC constructs and applied to single-cell data from human PCa specimens[34] showed enrichment of the signature of ARE activation primarily in normal luminal epithelial cells, while the signature of ARE repression was enriched in tumor luminal epithelium (Supplementary Fig. 11d–g). We next examined the impact of these signatures on the prognosis in patients with PCa. In a cohort of over 169,000 patients with clinically localized PCa[35] with available transcriptome profiles, the signature of ARE repression was associated with more aggressive tumors (Fig. 6b) and was highest in NCCN high and very high-risk disease (Fig. 6b), and higher Gleason grade, PSA (Supplementary Fig. 12a, b), and stage (Supplementary Tables 1 and 2), while ARE activation was associated with the opposite—less aggressive cancers (Fig. 6b, Supplementary Fig. 12c–e, and Supplementary Tables 3 and 4). ARE activation was associated with more luminal features, while the repressive signature was higher in tumors classified as basal subtype (Fig. 6b). In a cohort of 855 patients after radical prostatectomy[36], high expression of the ARE repressive activation signature was associated with worse metastasis-free survival (Fig. 6c), while the ARE activated signature higher expression was associated with improved metastasis-free survival (Fig. 6d). Finally, in cohorts of patients with metastatic castration-resistant PCa[37–40], the ARE activation signature was

associated with improved overall survival (Fig. 6e and Supplementary Fig. 12f), maintenance of luminal character (Fig. 6f) and responsiveness to enzalutamide (Supplementary Fig. 12g). Together, these data support that in both clinically localized and advanced, treatment-resistant human PCa, genes controlled by AREs are associated with prognosis and luminal versus basal phenotypes.

## Discussion

The transcription factor activity of AR is critical for the development, differentiation, and maintenance of the normal prostate, but "hijacking" of normal AR activity and reprogramming to drive oncogenesis is a fundamental feature of human PCa. The field has made major strides in defining the GAIN of function effects on the AR cistrome and transcriptional program associated with prostate tumorigenesis, including redistribution of AR to enhancers enriched for FOXA1 and HOXB13 motifs[3–6,20]. However, the functions of AR that are LOST during this process have been less explored. Here, based on the observation that canonical AREs are depleted in the AR cistromes of human cancers, we designed an experimental system to modulate genes specifically controlled by AREs in a genome-wide fashion.

We find that AREs are critical elements for defining distinct gene expression programs controlled by AR, and in particular, mediate differentiation-associated growth suppressive transcription in cells from a prostate lineage. AREs are considered to be a key part of all AR-directed transcription[3,21,41]—here, however, our data suggests these regulatory elements preferentially control growth suppressive, normal programs, while being relatively dispensable for oncogenic, proliferation-associated transcription.

The growth-suppressive nature of AR signaling in normal cells has been a well-known but poorly understood phenomenon at a mechanistic level[42,43]. While others have proposed that a transcriptional repression function of AR mediates this activity[44,45], our data show that direct activation of ARE-containing chromatin regions engages the growth-suppressive effects of AR and suggests that selective, context-dependent transactivation is responsible. Multiple epigenetic factors may play a role in coordinating such programs. Histone deacetylases[46] (HDACs), particularly class I HDACs like HDAC1, HDAC2, and HDAC3, remove acetyl groups from lysine residues on histones, engaging in an AR-centric transcriptional network in PCa[30]. Specifically, our data reveal that HDAC3 plays a unique role by binding to ARE-enriched chromatin regions, thus modulating the growth-suppressive transcription of AR signaling. This specificity not only highlights the potential of HDAC3 as a therapeutic target but also elucidates the mechanistic basis of AR/ARE's dependency on chromatin context and regulatory landscape.

The growth-suppressive effects of AR are also highly clinically relevant. We show here that the transcriptional signatures revealed by our ARE modulation strategy are associated with prognosis in human PCa patients, opening up new avenues for biomarker discovery and understanding both prognosis and response to agents targeting AR. Furthermore, activation of AR with supraphysiologic testosterone[47,48] or cycles of androgen stimulation/deprivation (bipolar androgen therapy) in PCa patients[49–52] has emerged as promising therapeutic strategies, with clinical trials in progress. This study provides mechanistic insight into these therapeutic approaches and potentially improves the selection of patients.

## Methods
### Cell lines
LNCaP cells (ATCC Item # CRL-1740) were cultured on poly-L-lysine (Sigma-Aldrich; cat. P1274) coated plates in RPMI-1640 containing 10% Fetal Bovine Serum (FBS) and incubated at 37 °C and 5% CO2. Cells were passaged twice weekly or when cultures reached 80% confluence. PC3 cells (ATCC CRL-1435) were maintained in DMEM supplemented with 10% FBS, while RWPE-1 cells (ATCC CRL-11609) were maintained in

Keratinocyte SFM (1X) medium supplemented with human recombinant epidermal growth factor (rEGF) and bovine pituitary extract. LAPC4 cells were a kind gift from Dr. Robert Reiter (UCLA) and grown in IMDM media containing 10% FBS incubated at 37 °C. LNCaP/AR cells were a kind gift from Dr. Charles L. Sawyers (MSKCC) and grown in RPMI-1640 media containing 10% FBS incubated at 37 °C. Prostate from the genetically normal mice was harvested at 2–3 months of age and processed and grown as 3D Matrigel culture as previously described[5,11]. All 2D and 3D cultures underwent monthly mycoplasma testing using a highly sensitive PCR-based kit (ABM; cat. G238). Cell line identity, when applicable, was confirmed annually through STR profiling provided by ATCC's cell authentication service.

### Generation of MACC constructs
The oligos used in the MACC construction are provided in Supplementary Table 5.

#### Cloning AR DBD domain and NLS with ERT2.
1. The AR DBD domain was cloned with a mutant estrogen ligand-binding domain (ERT2) to achieve tamoxifen inducibility. The AR DBD domain DNA fragment was amplified using Herculase II Fusion DNA Polymerase (Agilent Technologies; cat. 600677) with 36 cycles and annealing at 63.5 °C, then purified with the Qiagen PCR purification kit (Qiagen; cat. 28004).
2. To link the AR DBD domain with ERT2, the pRetroQ-Cre-ERT2 plasmid (Addgene #59701) and AR DBD domain DNA fragment were cut by NheI (N-terminal) and XhoI (C-terminal) and purified with the QIAquick Gel Extraction Kit (Qiagen; cat. 28704). After ligation with T4 DNA Ligase (New England; cat. M0202S), bacterial clone screening and sanger sequencing were performed to verify the AR DBD domain DNA fragment in frame and unmutated. This inducible construct (AR-DBD_ERT2 construct) will be further modified with chromatin structure modulation.

#### Cloning ARE with modulation of chromatin structure.
1. To modulate the chromatin structure around the AREs on our inducible construct, we linked repressive or activating chromatin modifying domains (H3K9 methyltransferase KRAB or transcriptional activator VP64) and an epitope tag (3xFlag tag) to the C-terminal of our AR-DBD_ERT2 construct. We amplified 3xFlag-KRAB (KRAB-dCas9 plasmid; Addgene #112195), 3xFlag-VP64 (pSL690 plasmid; Addgene #47753), and 3xFlag (KRAB-dCas9 plasmid; Addgene #112195) fragments using Herculase II Fusion DNA Polymerase (Agilent Technologies; cat. 600677) and primers with 30 cycles and annealing at 63 °C, purified them with the Qiagen PCR purification kit (Qiagen; cat. 28004) and verified all products by gel separation and sanger sequencing to ensure they were unmutated.
2. To add the 3xFlag-KRAB, 3xFlag-VP64, and 3xFlag fragments to the C-terminal of our AR-DBD_ERT2 construct, we used NheI-HF (New England Biolabs, cat. R3131L) to cut both the AR-DBD_ERT2 construct and the purified fragments, and Shrimp Alkaline Phosphatase (rSAP) (New England Biolabs, cat. M0371S) after AR-DBD_ERT2 construct NheI fragmentation to dephosphorylate. We then used T4 DNA Ligase (New England; cat. M0202S) to ligate the fragments with a 1:3 insertion to vector ratio at 16C overnight and transformed all reactions into Stbl3 chemically competent cells (Thermo Fisher Scientific; cat. C737303). We performed bacterial clone screening and Sanger sequencing to verify that they were in frame and unmutated.
3. These three inducible constructs (MACC constructs) will be further modified to enable lentiviral production.

#### Cloning the MACC constructs into the lentiviral and doxycycline-inducible backbone.
To facilitate lentiviral production, we transferred

all MACC constructs into a lentiviral backbone, specifically the pLenti PGK Blast V5-LUC (w528-1) plasmid (Addgene #19166). The process involved using SalI-HF (New England Biolabs, cat. R3138S) and AatII (New England Biolabs, cat. R0117L) to cleave both the MACC constructs and the backbone, followed by ligation, purification, and verification of all products through gel separation and Sanger sequencing to ensure their mutational integrity. To achieve the doxycycline-inducible ability, all the AR DBD domain with modulation of chromatin structure were cloned into a doxycycline-inducible lentiviral backbone, and verification of all products through gel separation and Sanger sequencing was conducted to ensure their mutational integrity. All plasmids were purified using the Midiprep kit (Zymo Research; cat. D4201).

### Generation of lentivirus
293T cells were cultured in 10 cm tissue culture plates until they reached 70–80% confluency. Transfection was performed using Lipofectamine 3000 Transfection Reagent (Thermo Fisher Scientific; cat. L3000015) with pMD2.G (lentiviral helper plasmid; Addgene #12259), psPAX (lentiviral helper plasmid; Addgene #12260), and the target transfer plasmid. Lentivirus was harvested 48/72 h after the start of transfection. The lentivirus supernatant from plates transfected with the same plasmid construct was pooled, and cellular debris was removed by filtration using Millipore's 0.45-μm filter unit. The filtered lentivirus supernatant was aliquoted and stored at −80 °C for later use.

### Generation of stable cell lines
To generate stable cell lines expressing the MACC construct, prostate 2D cells were infected with crude lentivirus at a ratio of 1:500 for 24 h. 3D organoid lines were generated using a spinoculation protocol at 650 g and 32C for 1 h. Subsequently, the cells and organoids were selected using Blasticidin (InvivoGen; cat. ant-bl-05) for 7 days. LNCaP cells were transfected with pLV[CRISPR]-hCas9:T2A:Puro plasmids, which contained sgRNAs specific either for control (pLV[CRISPR]-hCas9/Puro-U6>Scramble_gRNA1) or for HDAC3 (#1: CTACCTGGTTGATAACCGGC, #2: CCAGTCATCGCCTACGTTGA), purchased from VectorBuilder. For HDAC3 overexpression, LNCaP cells were transfected with pLV[Exp]-mCherry:T2A:Puro-EF1A > hHDAC3[NM_001355039.2] or empty vector from VectorBuilder. Following transfection, cells underwent selection with puromycin until resistant populations were established. These populations were then assayed for HDAC3 expression using immunoblot analysis.

### Xenograft tumor growth
6–8-week-old nude male mice were injected subcutaneously with 3 million LNCaP cells stably expressing the neutral, repressive, or activating MACC constructs with 100 μL of 1:1 Matrigel (Corning; cat. 354,234) and cells (resuspended in 1x PBS) and allowed to grow until they reached a volume of 300 mm³ on their flank. All mice were housed in the animal facility under conventional conditions with a light- (12 h dark/light circle), humidity- (30–70%) and temperature (70–74 °F)-controlled environment. Mice were fed with normal chow, and then switched to doxycycline-containing chow starting from week 7 until the study's termination. Tumor volume was measured weekly using electronic calipers, and total volume was calculated using the formula (wlh). The end of the experiments was determined after 3 weeks of Doxycycline chow, following the established protocol. If a humane endpoint was required, it was defined by parameters such as tumor size exceeding 1500 mm³, significant weight loss, or signs of distress in the mice. Mice were sacrificed if the tumor volume surpassed the predetermined upper limits, as outlined in the approved IACUC protocol.

### Animal studies approval
All mouse studies were approved by the Weill Cornell Medicine (WCM) Institutional Care and Use Committee under protocol 2015–0022.

### Immunoblot
For organoids, protein lysates were prepared after digestion of Matrigel (Corning; cat. 356231) with TrypLE Express Enzyme (Thermo Fisher Scientific; cat. 12605028) and washed in PBS and lysed in RIPA buffer supplemented with protease and phosphatase inhibitors. For cell lines, pelleted cells were washed in PBS and lysed in RIPA buffer supplemented with protease and phosphatase inhibitors. Proteins were quantified by BCA assay and separated on 4%–15% Protein Gels, protein was transferred to a nitro-cellulose membrane using the iBlot semi-dry system from Invitrogen, blocked in 5% milk in TBST, and incubated with primary antibody overnight rocking at 4 °C. Individual blots were washed 3x in TBST, incubated with species-specific HRP conjugated secondary antibody for 45 min at 24 °C, washed again 3x with TBST buffer and then imaged using the SuperSignal West Pico PLUS Chemiluminescent Substrate (Thermo Fisher Scientific; cat. 34578) on a ChemiDoc imaging system from BioRad. All antibodies and their concentrations used in this study can be found in Supplementary Table 6.

### Immunofluorescence
Cells were seeded onto poly-L-lysine-coated coverslips and cultured for 48–72 h before being fixed with 4% paraformaldehyde at room temperature for 15 min. After two PBS washes, cells were permeabilized with 0.1% Triton-X100 in PBS for 10 min. They were then washed again with PBS and blocked using 10% goat serum and 0.5% BSA in PBS for 30 min at room temperature. For organoids, immunofluorescence was performed following the previously described procedures[5,11]. Briefly, the organoids were suspended in Cell Recovery Solution (Corning; cat. 354253) to dissolve the Matrigel while maintaining the 3D cellular structure. Next, the organoids were harvested and embedded into fibrinogen-thrombin clots. Paraffin sections were processed at the Translational Research Program, Department of Pathology and Laboratory Medicine, WCM. Primary antibodies were diluted in blocking solution and applied overnight at 4 °C. The following day, cells were washed twice with PBS and incubated with the corresponding fluorescent secondary antibody, also in blocking solution, for 30 min in the dark. After three PBS washes, coverslips were mounted using Prolong Gold antifade mountant containing DAPI (Thermo Fisher Scientific; cat. P36931) and visualized under a fluorescent microscope.

### Growth curves
5000 cells per cell line were plated in biological triplicates and monitored for confluency changes over time using Incucyte software (2022B Rev1) for the specified duration. The mean confluency and standard error are plotted for each time point.

**Treatments.** For androgen stimulation experiments, 5000 cells were seeded into a 96-well plate containing phenol-red free RPMI-1640 supplemented with 5% charcoal dextran-treated serum (HyClone; cat. SH30068.03IR25-40). After allowing the cells to adhere overnight, they were treated with varying concentrations of DHT (dissolved in EtOH). Cell growth was monitored using Incucyte software, and the average of 4 images per well was plotted in biological triplicates for each cell line and condition.

For Tamoxifen (Tam) treatment, 5000 cells were seeded in a 96-well plate containing phenol-red free RPMI-1640 with 10% FBS. After an overnight attachment, cells were treated with varying concentrations of Tamoxifen (dissolved in EtOH). Confluency was tracked and calculated using Incucyte software. The average of four images per well was plotted in biological triplicates for each cell line and condition. When applicable, Caspase-3/7 Green Reagent for Apoptosis (Sartorius; cat. 4440) was added to the media as per the manufacturer's instructions, and the fluorescence signal was monitored with the Incucyte live-cell analysis system.

## RNA extraction and library preparation

LNCaP MACC cells and RWPE1 MACC cells were stimulated with either EtOH or Tamoxifen (Sigma-Aldrich; cat. T176) and harvested for RNA-seq in biological replicates. Total RNA was extracted, and DNaseI treated by RNeasy Mini Kit (Qiagen; cat. 74104). Nanodrop quantified RNA was checked by Bioanalyzer RNA 6000 Nano Kit (Agilent Technologies). Samples with RNA integrity number >10 were used for library preparation (Illumina Stranded mRNA Prep kit) and sequenced on Illumina NovaSeq 6000 at the WCM Genomics Core.

## RNA-seq analysis

Data was processed using (v3.10) of the nf-core collection of workflows[53]. The pipeline was executed with Nextflow[54] v22.10.4 with the following command:

*nextflow run nf-core/rnaseq -r 3.10 --input samplesheet.csv --genome GRCh37 -profile singularity.*

Briefly, raw FASTQ files were aligned to the GRCh37 (hg19) reference genome and quantified using salmon (v 1.9.0)[55]. Differential gene expression was performed in R with DESeq2 (1.28.0)[56] and iDEP[57]. Gene set enrichment analysis (GSEA)[58] was conducted in pre-ranked mode to identify enriched signatures from the Molecular Signature Database (MSigDB).

## ChIP and ChIP-sequencing

ChIP was performed as previously described[5]. LNCaP MACC cells were treated with 2.5 μM Tamoxifen for either 4 or 18 h. For each replicate, 20 million cells were fixed with 1% formaldehyde for 10 min at 24 °C, quenched with glycine, washed with PBS, and stored at −80 °C. The fixed pellets were lysed in SDS buffer and sonicated to obtain DNA fragments between 250-400 bp. Samples were incubated with Protein A-conjugated beads and antibodies overnight at 4 °C, washed with increasing salt buffers, and DNA was eluted at 65 °C overnight. All samples were treated with RNase A for 30 min at 37 °C, followed by Proteinase K treatment at 65 °C for 1 h. DNA was purified using phenol chloroform, and individual ChIP samples were validated by qPCR. ChIP-seq libraries were prepared using the KAPA Hyper Prep Kit (Roche; cat. 08278539001) with 20 ng DNA per sample, following the manufacturer's protocol. Library quality, purity, and size were assessed using DNA High Sensitivity Bioanalyzer chips (Agilent; cat. 5067-4626). Libraries passing quality control were quantified with the Library Quantification Kit (Roche, cat. 07960298001). Pooled libraries were sequenced on a NovaSeq 6000 at the WCM Genomics Core. ChIP-seq utilized ERalpha (ERa) antibody (Santa Cruz Biotechnology; cat. sc-8002, AB_627558) to generate MACC peaks, along with H3K27ac antibody (Abcam; cat. ab4729, AB_2118291).

## ChIP-seq data analysis

Briefly, the quality of the raw reads (FASTQ files) was validated using FastQC software (Version 0.11.7), and single-end reads with a score > 29 were aligned to the hg19 human reference genome using Bowtie2 software (v2.2.9)[59] with default parameters. The resulting SAM files were converted to BAM format, sorted, PCR duplicates removed, ENCODE blacklist regions eliminated, and final BAM files indexed using Samtools (v1.7)[60]. Replicate BAM files were then combined to generate RPKM-normalized bigwig files for each factor using Deeptools (v3.0)[61]. These bigwig files were used to create heatmaps and binding profiles using Deeptools v3.0 (computematrix, plotProfile, and plotHeatmap functions).

## Peak calling

Peaks were called using MACS2[62] with a *p* value < $10^{-8}$ or *q* = 0.05 (replicates combined) using the narrow peak caller and matched input as background. Peak overlap and Venn diagrams were generated using pybedtools and bedtools intersect function[63,64] and were defined as overlap (more than or equal to) 1 bp. Where indicated, parental AR ChIP-seq data was utilized from GSE117430[5] and processed and analyzed as above.

A conserved MACC peak set was defined as a peak shared among 2 or more datasets (Neutral, Repressive, or Activating). Overlap between all conditions was determined using bedtools (v2.28.0) intersect function with a minimum overlap of 1 bp. Subsets of these peaks were generated using bedtools (v2.28.0) -subtract function.

**Motif analysis.** Motif analysis was performed using Homer software (v4.8.3)[65] by analyzing a 200 bp window around the center of each peak. Motif density around peaks was calculated using Homer and JASPER[66] definitions of the conserved motif. To determine motif enrichment between datasets with similar peak numbers, peak sets of the control were used as background (-bg flag in findmotifsgenome.pl function). A p-value less than or equal to $10^{-20}$ was considered significant for motif enrichment, unless otherwise indicated.

Analyses of AR peaks containing AREs conducted from published ChIP-seq were conducted using peak sets in their published form and the JASPER definition of an ARE. Peaks were assessed for the presence of this motif using HOMER's annotatepeaks.pl -m function (v3.0) within a 100 bp window from the peak center.

The density of specific motifs around a peak set was conducted using the annotatepeaks.pl function in HOMER v3.0 with a window of 2,400 bp from the peak center binned every 10 bp. Comparison among ChIP conditions was done by subtracting the determined motif frequency/bp/peak from the other. Signal above 0 was considered enrichment and below 0 depletion. These profiles were plotted in PRISM v9.2.0 and traces were generated using the smooth, differentiate or integrate curve function with $2^{nd}$ degree curve smoothing.

H3K27ac ChIP-seq data from human PCa or normal tissue was downloaded from GSE130408 and analyzed in its published form[20]. The HDACs and EZH2 ChIP-seq data was downloaded from GSE28950 and analyzed in its published form[30]. Principal component analyses of these samples using conserved MACC ChIP-seq peaks were conducted using DeepTools (v3.0)[61] MultiBigWigSummary and PlotPCa packages.

## ChromHMM

ChromHMM[25] was employed to delineate chromatin states, utilizing six distinct histone marks (H3K4me1, H3K4me2, H3K4me3, H3K27me3, H3K36me3, H3K79me2) in the presence of androgen stimulation, along with MACC H3K27ac histone mark data, with default parameters for the hg19 genome. The histone mark data (in bed file format) were obtained from ChIP-Atlas[67], adhering to the hg19 genome version, and applying a MACS2 cutoff of q < 1E-05, under the identifiers SRX5060896 (H3K4me1), SRX8142314 (H3K4me2), SRX4411668 (H3K4me3), SRX4411671 (H3K27me3), SRX120296 (H3K36me3), and SRX8142325 (H3K79me2). A 6-state model (E1–E6) was chosen and implemented based on histone mark enrichments, in line with methodologies previously described by ENCODE and the Roadmap Project. For each state, considering the composition of the histone marks and their association with genomic features, such as laminB1 LADs and CpG island features, a numeric transformation was applied to the categorical states. This transformation involved assigning numeric values to the six chromatin states: heterochromatin/ZNF/repeats (E1), quiescent/no mark (E2), quiescent/weak repression (E3), repressed chromatin (E4), bivalent enhancers and promoters (E5), and active promoters and enhancers (E6).

## Gene signatures

Differential expression analyses were performed between tamoxifen and control treatments in each cell (LNCaP MACC cells and RWPE1 MACC cells), and the significantly overexpressed and underexpressed genes were defined as neutral, repressive, and activating signatures. The signature scores were defined as the sum of z-scores from overexpressed genes and underexpressed genes for each signature. Signature score = sum (z-scores from overexpressed genes)−sum (z-scores from underexpressed genes).

Gene signatures for AR score, cell cycle, neuroendocrine (NE) differentiation, luminal and basal phenotypes, stem-like, and epithelial-mesenchymal transition (EMT) were defined based on well-established gene sets in the literature[68,69]. Signature Score = sum (z-scores of genes in the tamoxifen-treated group)−sum (z-scores of genes in the control group). The resulting enrichment scores were then used for visualization.

## Human data analysis

Human prostate scRNA-seq was downloaded from GSE120716[34] and analyzed in its published form.

All human data in this study were collected in accordance with the International Ethical Guidelines for Biomedical Research Involving Human Subjects. We confirm that informed consent was obtained from all participants prior to data collection.

De-identified gene expression profiles were obtained prospectively from clinical usage of the Decipher prostate genomic classifier between January 2016 and June 2023, n = 169,123 (Veracyte Inc, San Diego, California). Samples were obtained from either prostate biopsy or radical prostatectomy, and ordering criteria for the genomic classifier exclude prior treatment with hormone therapies or radiotherapy. All tumors were prospectively gathered into the Decipher Genomics Resource for Intelligent Discovery (GRID) database (NCT02609269)[35]. A retrospective cohort from individual patient-level data generated in a prior meta-analysis with long-term follow-up (META855; n = 855)[36] was used to test associations with time to metastasis after radical prostatectomy. Time-to-event end points were shown graphically using the Kaplan−Meier method. Multivariable Cox regressions were used to compare time to failures. The statistical significance of differences in continuous and categorical variables between groups was assessed using Kruskal−Wallis and Pearson $X^2$ tests, respectively. Given the exploratory nature, no adjustments for multiple hypothesis testing were performed, all tests were two-sided, and all analyses were performed using R version 4.0.3.

The CRPC cohort consisted of 123 biopsies from male metastatic castration-resistant prostate cancer (mCRPC) patients, derived from two studies[37,38], with diverse clinical characteristics and treatment histories. Other mCRPC cohorts[39,40] were also followed using the same strategies with the original published format. Baseline biopsy samples captured initial gene expression at first biopsy after mCRPC diagnosis. RNA-seq aligned with STAR provided gene expression data from gencode v.28, normalized to TPM, and log2 transformed. Z-scores standardized expression values per gene were calculated for ARE activating (VP64) and ARE repressed (KRAB) gene expression scores, summed log2 fold changes for positively and negatively expressed genes. The primary endpoint was overall survival from biopsy to death/last follow-up. Patients were stratified by quartiles of VP64 and KRAB scores. Kaplan−Meier curves visualized survival analysis, and Cox proportional hazards regression evaluated gene expression's impact on survival as continuous variables per quartile. Basal/luminal gene expression score was calculated as previously described[38] and was correlated with VP64 and KRAB scores, with Spearman analysis of correlation.

## Statistics and reproducibility

No statistical method was used to predetermine the sample size. All experiments were replicated at least twice. No data were excluded from the in vivo analysis. For in vitro studies, standard deviation (SD) is reported in the figure legends for technical replicates from representative experiments performed in duplicates or triplicates. Statistical significance was determined as indicated in the figure legends.

## Reporting summary

Further information on research design is available in the Nature Portfolio Reporting Summary linked to this article.

## Data availability

All raw next-generation sequencing, ChIP and RNA−seq data generated in this study have been deposited in the Gene Expression Omnibus (GEO) repository at NCBI under accession code GSE231516. The parental AR ChIP-seq data have been previously published[5] and are available at the GEO (GSE117430). The H3K27ac ChIP seq data from human PCa or normal tissue have been previously published[20] and are available at the GEO (GSE130408). The RNA-seq data from TCGA primary prostate cancer patients have been previously published[32] and are available at the cBioPortal[70] for Cancer Genomics. The human prostate scRNA-seq data have been previously published[34] and are available at the GEO (GSE120716). The AR ChIP-seq data of FOXA1 manipulation have been previously published[14,15] and are available at the GEO (GSE30623) and E-MTAB-1749. The LNCaP RNA-seq data have been previously published[12] and are available at the GEO (GSE153585). The HDACs and EZH2 ChIP-seq data have been previously published[30] and are available at the GEO (GSE28950). Source data are provided with this paper. All the other data are available within the article and its Supplementary Information.

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

## Acknowledgements

We are indebted to prostate cancer patients and families who contributed to this research. We thank Dr. Dawid Nowak, Dr. Pengbo Zhou, and Dr. Jonathan E. Shoag (WCM) for helpful discussion. We thank the Weill Cornell Medicine (WCM) Genomics Core Facility, the Biospecimen and Pathology Core, and the Computational and Biostatistics Core of the WCM SPORE in Prostate Cancer and Memorial Sloan Kettering Cancer Center cBioPortal. The authors wish to acknowledge the support provided by the Freiburg Galaxy Team for the Galaxy server (https://usegalaxy.eu). This work was funded by: the NCI, US (WCM SPORE in Prostate Cancer, P50CA211024, R37CA215040, and R01CA233650, C.E.B.), Damon Runyon Cancer Research Foundation, MetLife Foundation Family Clinical Investigator Award, US (to C.E.B.), and the Prostate Cancer Foundation, US (M.A.A. and D.L.).

## Author contributions

M.A.A., X.C., and C.E.B. designed the study. M.A.A., X.C., and K.L. designed and executed experiments. D.L., U.C., and A.S. designed and executed bioinformatic analyses. A.H., Y.L., and E.D. performed analysis in clinically localized prostate cancer cohorts. M.S. and F.Y.F. performed analysis in CRPC cohorts. F.K., M.L., and B.D.R. provided specimens and performed pathologic analysis. K.L., L.D.D., J.T.G., J.O., and J.W. provided technical support and performed animal experiments. M.A.A., X.C., D.L., and C.E.B. wrote the manuscript. All authors participated in the critical evaluation and revision of the manuscript.

## Competing interests

M.S. reports grants from the Swedish Research Council, the Swedish Society of Medicine, and the Prostate Cancer Foundation during the conduct of the study. A.H., Y.L., and E.D. are employees of Veracyte, Inc. M.A.A. and D.L. are currently employees of Loxo Oncology. C.E.B. is a co-inventor on a patent issued to Weill Medical College of Cornell University on SPOP mutations in prostate cancer. F.Y.F. reports fees from Janssen Oncology, Bayer, PFS Genomics, Myovant Sciences, Roivant Sciences, Astellas Pharma, Foundation Medicine, Varian, Bristol Myers Squibb (BMS), Exact Sciences, BlueStar Genomics, Novartis, and Tempus; other support from Serimmune and Artera outside the submitted work. The authors declare no other potential competing interests.
