## [Transparent Peer Review file · Nature Communications]

Canonical androgen response element motifs are tumor suppressive regulatory elements in the prostate

Corresponding Author: Dr Christopher Barbieri

This manuscript has been previously reviewed at another journal. This document only contains reviewer comments, rebuttal and decision letters for versions considered at Nature Communications.

Version 0:

Reviewer comments:

Reviewer #1

(Remarks to the Author)

I have previously reviewed an earlier version of this manuscript. The new version has incorporated a significant amount of new data, which clearly strengthens the main conclusion and the overall impact of the work. My previous concerns have been sufficiently addressed as well. Therefore, I recommend the acceptance of this revised manuscript.

--Ping Mu

Reviewer #2

(Remarks to the Author)

The manuscript has been substantially improved. Most of my initial concerns were addressed. The new data showing HDAC3 function is interesting. It would be nice if HDAC3 ChIPseq was done in MACC Activating model, rather than the less relevant DHT mode, to see if it is specific recruited to those activated ARE sites. However this is optional.

Reviewer 1 Comment:

General comments:

“I have previously reviewed an earlier version of this manuscript. The new version has incorporated a significant amount of new data, which clearly strengthens the main conclusion and the overall impact of the work. My previous concerns have been sufficiently addressed as well. Therefore, I recommend the acceptance of this revised manuscript.”

Reviewer 2 Comment:

General comments:

“The manuscript has been substantially improved. Most of my initial concerns were addressed. The new data showing HDAC3 function is interesting. It would be nice if HDAC3 ChIPseq was done in MACC Activating model, rather than the less relevant DHT mode, to see if it is specific recruited to those activated ARE sites. However this is optional.”

Response: We thank the reviewers for their positive comments and appreciate that the manuscript was improved thanks to their feedback and suggestions.